# Effectiveness of virtual reality technology in rehabilitation after anterior cruciate ligament reconstruction: A systematic review and meta-analysis

Yunchuan Li[1,◉], Junjie Peng[1,◉], Jintao Cao[1], Yang Ou[1], Jiaming Wu[1], Weisha Ma[2], Feng'e Qian[3*], Xiaoqian Li[4*]

1 School of Nursing, Yunnan University of Traditional Chinese Medicine, Kunming, Yunnan, China,
2 Chuxiong Yi Autonomous Prefecture Hospital of Traditional Chinese Medicine, Chuxiong, Yunnan, China,
3 Department of Nursing, Yunnan University of Traditional Chinese Medicine, Kunming, Yunnan, China,
4 The People's Hospital of Chuxiong Yi Autonomous Prefecture, Chuxiong, Yunnan, China

◉ These authors contributed equally to this work.
* FengeQian@outlook.com (FQ); 2751209196@qq.com (XL)

## Abstract

### Background

Anterior cruciate ligament reconstruction (ACLR) can be fully recovered with effective rehabilitation, which also lowers the risk of developing osteoarthritis in the knee. Virtual reality technology (VRT) has been used for rehabilitation after ACLR. However, it is unclear how VRT compares to traditional therapy in terms of effectiveness.

### Design

A systematic review and a meta-analysis.

### Objectives

We hypothesised that VRT would be a more effective treatment than traditional therapy in post-ACLR rehabilitation. This study aimed to evaluate the effects of VRT on rehabilitation following ACLR, providing insights for its application in clinical settings.

### Materials and methods

A systematic review and meta-analysis of randomized controlled trials (RCTs) was performed using RevMan and Stata software according to PRISMA guidelines. We conducted a systematic search of the PubMed, Web of Science, Embase, The Cochrane Library, EBSCO, CNKI, CBM, VIP, and Wanfang databases for RCTs examining the effects of VRT in patients following ACLR. The literature search was conducted from the inception of the database to March 2024, utilizing keywords such as "anterior cruciate ligament," "anterior cruciate ligament reconstruction," "anterior cruciate ligament injury," and "virtual reality." The outcome indicators comprised knee function, walking function, gait function, and knee muscle strength. We assessed the quality of RCTs using the Cochrane Risk of Bias tool and the Jadad scale.

**Data availability statement:** All relevant data are within the manuscript and its Supporting Information files.

**Funding:** This study was supported by Science and Technology Program of Yunnan Provincial Department of Science and Technology (Grant NO. 202301AZ070001-052), Yunnan Provincial Department of Education Science Research Fund Project (Grant NO. 2023Y0493). The funders had no role in study design, data collection and analysis, decision to publish, or preparation of the manuscript. The authors have no conflicts of interest to declare.

**Competing interests:** The authors have declared that no competing interests exist.

## Results

There were a total of 6 RCTs included in this study, involving 387 patients who had undergone ACLR. The experimental group comprised 194 patients, while the control group comprised 193 patients. The findings demonstrated that VRT significantly enhanced knee function, walking ability, gait function, and knee muscle strength post-ACLR. Specifically, it led to improvements in the IKDC score (MD: 4.23; 95% CI 1.76-6.71), FAC score (0.40; 0.32-0.48), Lysholm score (6.36; 3.05-9.67), step length (3.99; 2.72-5.27), step speed (0.13; 0.10-0.16), step frequency (4.85; 0.22-9.47), extensor peak torque (12.03; 3.28-20.78), and flexor peak torque (14.57; 9.52-19.63). Subgroup analysis revealed that fully immersive VR did not significantly improve knee function as compared to non-immersive VR.

## Conclusion

This study is the first to systematically compare VRT with traditional therapy, and we found that VRT is a more effective treatment than traditional therapy in post-ACLR rehabilitation. This provides evidence for integrating VRT into post-ACLR rehabilitation protocols. However, more high-quality studies with large samples are needed to verify the findings.

## Protocol registration

This study has been registered in PROSPERO (No. CRD42024534918).

## Introduction

Anterior cruciate ligament (ACL) rupture is a common and devastating injury [1], with over 2 million cases reported globally each year [2]. Anterior cruciate ligament reconstruction (ACLR) surgery is largely regarded as the most successful means of restoring patients with ACL rupture to their pre-injury range of motion [3,4]. However, surgical knee dysfunction, decreased muscle strength, and proprioceptive impairments persist [5–7]. Furthermore, studies have shown that nearly 80% of patients develop osteoarthritis of the knee following ACLR [8], which is the leading cause of knee pain, decreased function, and disability [9]. As a result, it is necessary to develop a safe and effective rehabilitation program to recover knee function and lower the risk of postoperative knee osteoarthritis. Current research has demonstrated the benefits of neuromuscular electrical stimulation, centrifugal training, and virtual reality (VR) in postoperative rehabilitation following ACLR [10–12]. Among these, VR is a new technology that is gaining popularity because of its intriguing, inventive, and safe qualities. VR is a digital simulation of a computer-generated situation or environment that generates a realistic environment for task-specific training in which the user can orientate and interact in 3D through multiple sensory modalities [13,14]. Previous meta-analysis have shown virtual reality technology (VRT) as an effective intervention for improving upper limb motor function in stroke patients [15], reducing fear of falling in multiple sclerosis patients [16], and improving mobility in Parkinson's patients [17]. However, to the best of our knowledge, no meta-analysis has been performed to critically evaluate the efficacy of VRT as an intervention for rehabilitation after ACLR.

Currently, although several studies [18–20] have shown the beneficial effects of VRT in rehabilitation following ACLR, the findings are not entirely uniform, with some results showing significant improvement compared to conventional therapy and some showing no difference. In addition, there is a lack of consensus regarding the use of VRT in ACLR

rehabilitation. Therefore, clarifying whether and to what extent VRT is effective for rehabilitation after ACLR is important for future research and patient treatment choice.

Based on this background, we hypothesised that a systematic review and meta-analysis of randomized controlled trials (RCTs) would provide sufficient scientific evidence to consider VRT as a more effective therapy than traditional therapy in rehabilitation after ACLR. Therefore, the aim of this study was to comprehensively evaluate the efficacy of VRT compared with conventional therapies in rehabilitation after ACLR, which will provide a basis for decision-making on its clinical application and evidence for consensus building.

## Methods

### Design

A systematic review and a meta-analysis.

### Protocol and registration

This systematic review and meta-analysis followed the PRISMA recommendations [21] (S1 Checklist). The study is registered in the PROSPERO database with the number CRD42024534918.

### Search strategy

We systematically searched PubMed, Web of Science, The Cochrane Library, Embase, EBSCO, CNKI, CBM, VIP, and Wanfang databases using a combination of MeSH terms and free text. Ongoing and unpublished trials were also searched through the Chinese Clinical Trial Registry and Clinical Trials.gov. We searched for the following terms: ("Anterior Cruciate Ligament"[Mesh] OR "Anterior Cruciate Ligament Reconstruction" OR "Anterior Cruciate Ligament Injury" OR "Cruciate Ligament, Anterior" OR "Anterior Cruciate Ligaments" OR "Cruciate Ligaments, Anterior" OR "Ligament, Anterior Cruciate" OR "Ligaments, Anterior Cruciate" OR "Anterior Cranial Cruciate Ligament" OR "Cranial Cruciate Ligament" OR "Cranial Cruciate Ligaments" OR "Cruciate Ligament, Cranial" OR "Cruciate Ligaments, Cranial" OR "Ligament, Cranial Cruciate" OR "Ligaments, Cranial Cruciate" OR "ACL" OR "ACLR") AND ("Virtual Reality"[Mesh] OR "Reality, Virtual" OR "VR" OR "Virtual Environment" OR "Virtual Rehabilitation" OR "Immersive Multimedia" OR "Computer-simulated Reality" OR "Video Game" OR "Virtual Game" OR "Virtual Therapy"). Two autonomous researchers conducted searches without any intervention. The literature was searched from inception to March 2024. All search strategies (S1 File) are provided as supplementary material.

### Inclusion and exclusion criteria

According to the PICOS methodology, inclusion criteria for selecting appropriate studies were determined. (1) Population: patients who have received ACLR without restriction on sex, age, or graft type. (2) Intervention: any VRT intervention. (3) Comparison/control: conventional rehabilitation or any other intervention not involving VRT. (4) Outcomes: knee function, walking function, gait function, and knee strength. (5) Study design: RCTs. Studies meeting the following criteria were excluded: reviews, conferences, abstracts, case reports, duplicate reports, grey literature, non-RCTs, and lack of data required for meta-analysis.

### Study selection and data extraction

Two researchers independently screened and extracted the literature from different databases. During the literature screening, the initial selection was based on the title and abstract, and

any irrelevant material was discarded. Studies that did not meet the criteria were excluded after reading the entire article. A third researcher was included in the decision-making process if there was any disagreement. The following data were extracted from the included studies: first author, year, sample size, study population, study design, baseline characteristics of participants, interventions, control measures, intervention frequency, immersion level, outcome indicators, and measurement tools.

## Quality appraisal

The Cochrane Risk Assessment Scale [22] and the Jadad Scale [23] were used to grade and score the methodological quality of the included literature respectively. The Cochrane risk assessment scale includes the following 7 items: (1) Random sequence generation; (2) Allocation concealment; (3) Blinding of participants and p1ersonnel; (4) Blinding of outcome assessment; (5) Incomplete outcome data; (6) Selective reporting; (7) Other bias. The results are judged as "low risk," "high risk," or "unclear" according to the standards. The Jadad scale comprises three sections that evaluate study randomization, double-blinding, and withdrawal/withdrawal, with scores ranging from 0 to 5. A study is considered to be of acceptable quality if the score is equal to or greater than 3, while a score below 3 indicates that the study is of low quality.

## Data synthesis

Meta-analysis was performed using RevMan5.3 software. The included outcome indicators were continuous variables with mean difference (MD) and 95% confidence intervals (CI). Heterogeneity was analyzed using the $\chi^2$ test combined with the $I^2$ value. When $I^2 < 50$ and $P \geq 0.1$, it indicated that there was no heterogeneity among the studies, and the fixed-effects model was used for combined analysis. When $I^2 \geq 50\%$ and $P < 0.1$, it indicated that there was heterogeneity among the studies, and the random effects model was used for analysis. When heterogeneity was found, sensitivity analysis was used. By excluding literature one by one, the source of heterogeneity was explored and the stability of the results was tested. We performed subgroup analyses based on a range of designs and variables, analyzing subgroups in terms of immersion level and intervention start time. The Egger's test was used to analyze publication bias using Stata15.1 software. $P < 0.05$ means the difference is statistically significant.

# Results

## Literature selection

The initial search produced a total of 409 scholarly articles, consisting of 328 articles in English and 81 articles in Chinese. EndNoteX9.1 was utilized to eliminate 160 redundant entries, resulting in a remaining total of 269 articles. After reviewing the title and abstract of the articles, 251 pieces of literature that seemed unrelated were eliminated, leaving only 18 pieces of literature. After carefully examining the content of the articles, we eliminated 8 papers that did not meet the criteria of being randomized controlled studies, lacked the necessary outcome indicators, were not available in full text, or had incomplete data. As a result, we were left with 10 papers. After further screening, 4 studies with inconsistent outcome metrics were excluded, and finally a total of 6 RCTs were included [24–29]. Fig 1 depicts the process of literature screening.

## Study characteristics

Following a rigorous selection process, 6 RCTs involving a total of 387 patients who underwent ACLR were included [24–29]. The VR group consisted of 194 patients, whereas the

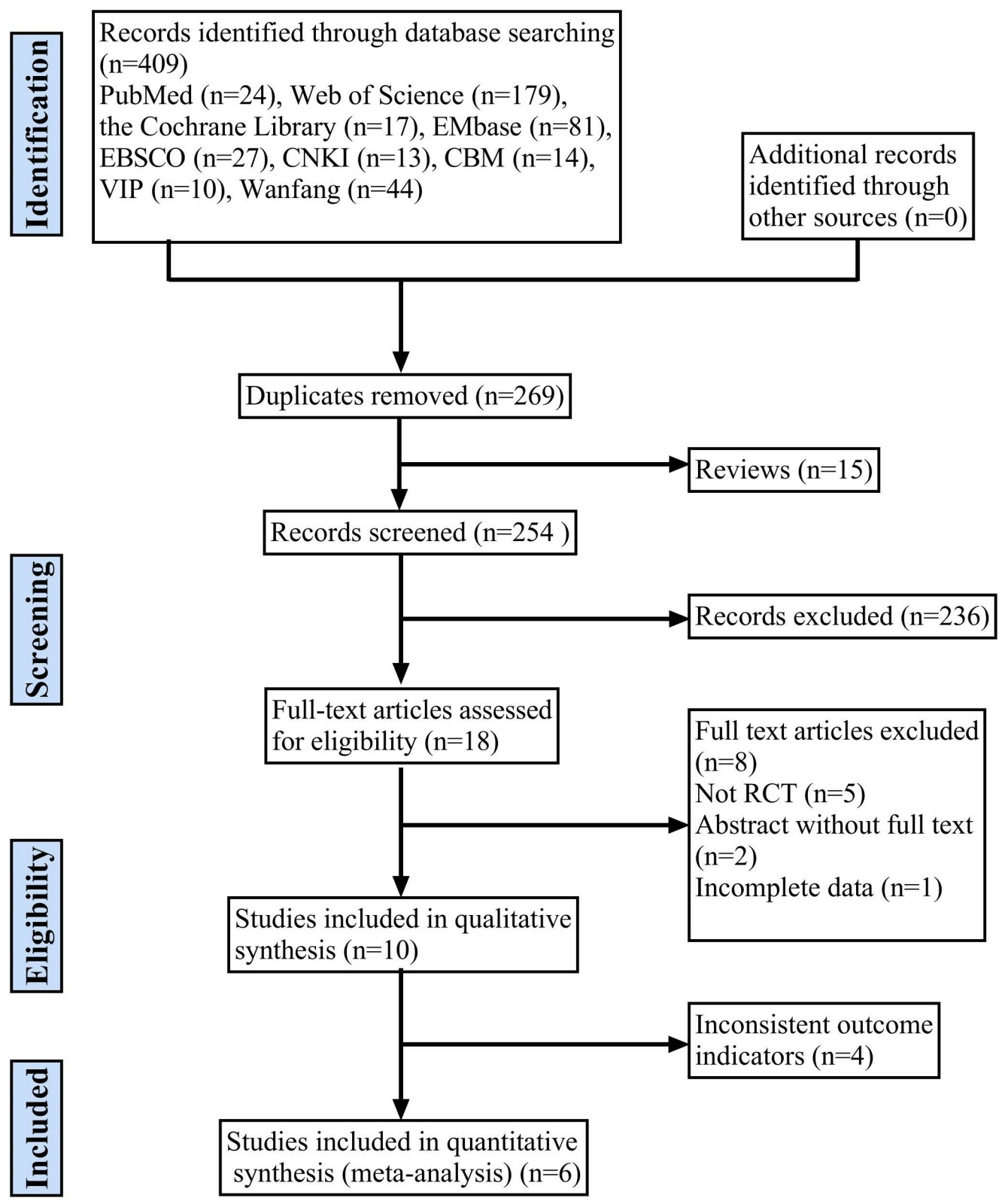

**Fig 1. Flow chart of the search for eligible RCTs.**

control group had 193 patients. Five studies included IKDC scores [24–26,28,29], four studies included FAC scores [25,27–29], two studies included Lysholm scores [26,27], two studies included stride length, stride speed, and stride frequency [27,28], in addition, two studies included the extensor peak torque (EPT) along with flexor peak torque (FPT) [25,29]. Refer to Table 1 for specifics.

## Quality evaluation

Six articles [24–29] discuss ways for generating random sequences, including the random number table method and the computer random sequence method. None of the 6 studies provided detailed descriptions of allocation concealment, making it unclear if allocation concealment was carried out. None of the 6 studies were blinded. There was a study in which 1 patient in the control group was disengaged due to early discharge [28], which did not have a serious impact on the outcome of the study. Six studies received a score of 3 on the Jadad scale [24–29], as outlined in Table 1, Figs 2 and 3. The Cochrane risk-of-bias tool for randomized trials (S3 Table) is provided as supplementary material.

**Table 1. Characteristics of included studies.**

| Control | Intervention start time | Frequency of intervention | Intervention cycle | VR equipment | Immersion level | Outcomes | Jadad score |
|---|---|---|---|---|---|---|---|
| conventional rehabilitation | 12th week | biweekly | 3 months | PlayStation VR headgear | immersive | (1) | 3 |
| exercise training | 1st week | 5 days/week, 2 times/day | 8 weeks | VR intelligent treadmill system | non-immersive | (1)(2)(7)(8) | 3 |
| conventional rehabilitation | 1st week | 5 times/week | 8 weeks | Dynstable VR 3D balance training system | non-immersive | (1)(3) | 3 |
| conventional rehabilitation | 5th week | 5 times/week | 8 weeks | VR system | non-immersive | (2)(3)(4)(5)(6) | 3 |
| conventional rehabilitation + balance training | 9th week | 5 days/week, 2 times/day | 3 months | Dynstable VR 3D balance training system | non-immersive | (1)(2)(4)(5)(6) | 3 |
| conventional rehabilitation | 6th week | 5 days/week, 2 times/day | 8 weeks | VR intelligent treadmill system + Dynstable VR 3D balance training system | immersive | (1)(2)(7)(8) | 3 |

| Study ID | Country | Sample size(T/C) | Sample source | Male and female | Age (Year, T/C) | Intervention |
|---|---|---|---|---|---|---|
| Gsangaya et al., 2023 [24] | Malaysia | 15/15 | Hospital | 23/7 | 28.6/25.1 | conventional rehabilitation + VR rehabilitation |
| Jin et al., 2022 [25] | China | 56/56 | Hospital | 55/57 | 38.32 ± 8.6/40.0 ± 9.35 | exercise training + VR balance training |
| Li et al., 2022 [26] | China | 50/50 | Hospital | 61/39 | 39.41 ± 4.64/40.12 ± 4.55 | conventional rehabilitation + VR balance training |
| Lin et al., 2022 [27] | China | 20/20 | Hospital | 23/17 | 29.74 ±5.72/28.51 ± 6.43 | conventional rehabilitation + VR training + Rehabilitation robot |
| Shi et al., 2021a [28] | China | 28/27 | Hospital | 39/16 | 27.7 ± 7.4/ 27.9 ± 7.5 | conventional rehabilitation + VR balance training |
| Shi et al., 2021b [29] | China | 25/25 | Hospital | 29/21 | 27.24 ± 5.81/27.92 ± 6.14 | immersive VR training |

T, treatment group; C, control group; (1) IKDC, international knee documentation committee; (2) FAC, functional ambulation classification; (3) Lysholm; (4) Step length; (5) Step speed; (6) Step frequency; (7) EPT, extensor peak torque; (8) FPT, flexor peak torque.

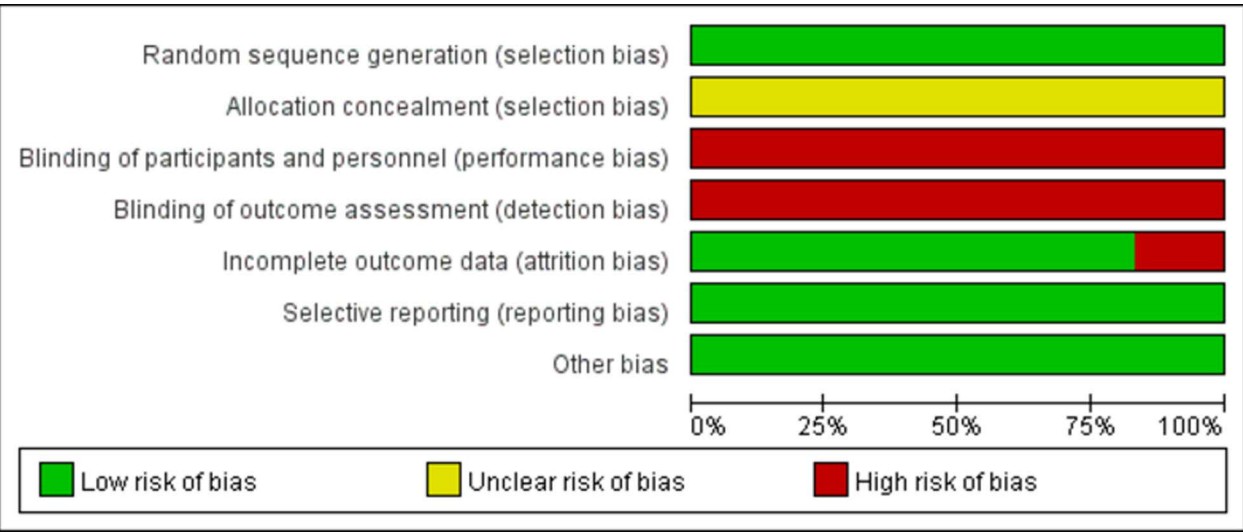

**Fig 2. Risk of bias graph.**

## Meta-analysis results

**IKDC score.** Five studies [24–26,28,29] utilized the IKDC assessment to evaluate knee function in a total of 347 patients. The studies included in the analysis exhibited significant heterogeneity ($\chi^2 = 36.09$, $P < 0.00001$, $I^2 = 89\%$), which led to the selection of a random-effects model for the meta-analysis. The results indicated a notable disparity in the IKDC scores between the VRT and control groups, with the former displaying higher scores. This discrepancy was found to be statistically significant (MD: 4.23; 95% CI 1.76-6.71, $P < 0.01$), as illustrated in Fig 4.

**FAC score.** Four studies [25,27–29] evaluated the walking function of a total of 257 patients using FAC. The studies showed no variation ($\chi^2 = 0.84$, $P = 0.84$, $I^2 = 0\%$), so we used a fixed-effects model to combine the analyses. The results of the meta-analysis revealed that the VRT group had a significantly better improvement in walking function compared to the control group (MD: 0.40; 95% CI 0.32-0.48, $P < 0.00001$), as depicted in Fig 5.

**Lysholm score.** Two studies [26,27] evaluated patient knee function using the Lysholm assessment, involving 140 patients. There was variation among the studies, so a random effects model was selected for meta-analysis. The results indicate a significant difference in Lysholm scores between the two groups (MD: 6.36; 95% CI 3.05-9.67, $P < 0.01$), as shown in Fig 6.

**Step length.** Two studies [27,28] examined the impact of VRT on step length in 95 individuals receiving ACLR. No heterogeneity was found among the studies ($\chi^2 = 1.18$, $P = 0.28$, $I^2 = 15\%$), hence a fixed-effects model was used to combine the analyses. The meta-analysis revealed a statistically significant improvement in step length in the VRT group compared to the control group (MD: 3.99; 95% CI 2.72-5.27, $P < 0.00001$), as depicted in Fig 7.

**Step speed.** Two studies [27,28] examined the impact of VRT on step speed in 95 patients who were undergoing ACLR. The studies showed no variation ($\chi^2 = 0.35$, $P = 0.55$, $I^2 = 0\%$), so a fixed-effects model was used to combine the analyses. Based on the results of the meta-analysis, it was found that the VRT group showed a significant improvement in step speed compared to the control group. The difference was statistically significant (MD: 0.13; 95% CI 0.10-0.16, $P < 0.00001$), as depicted in Fig 8.

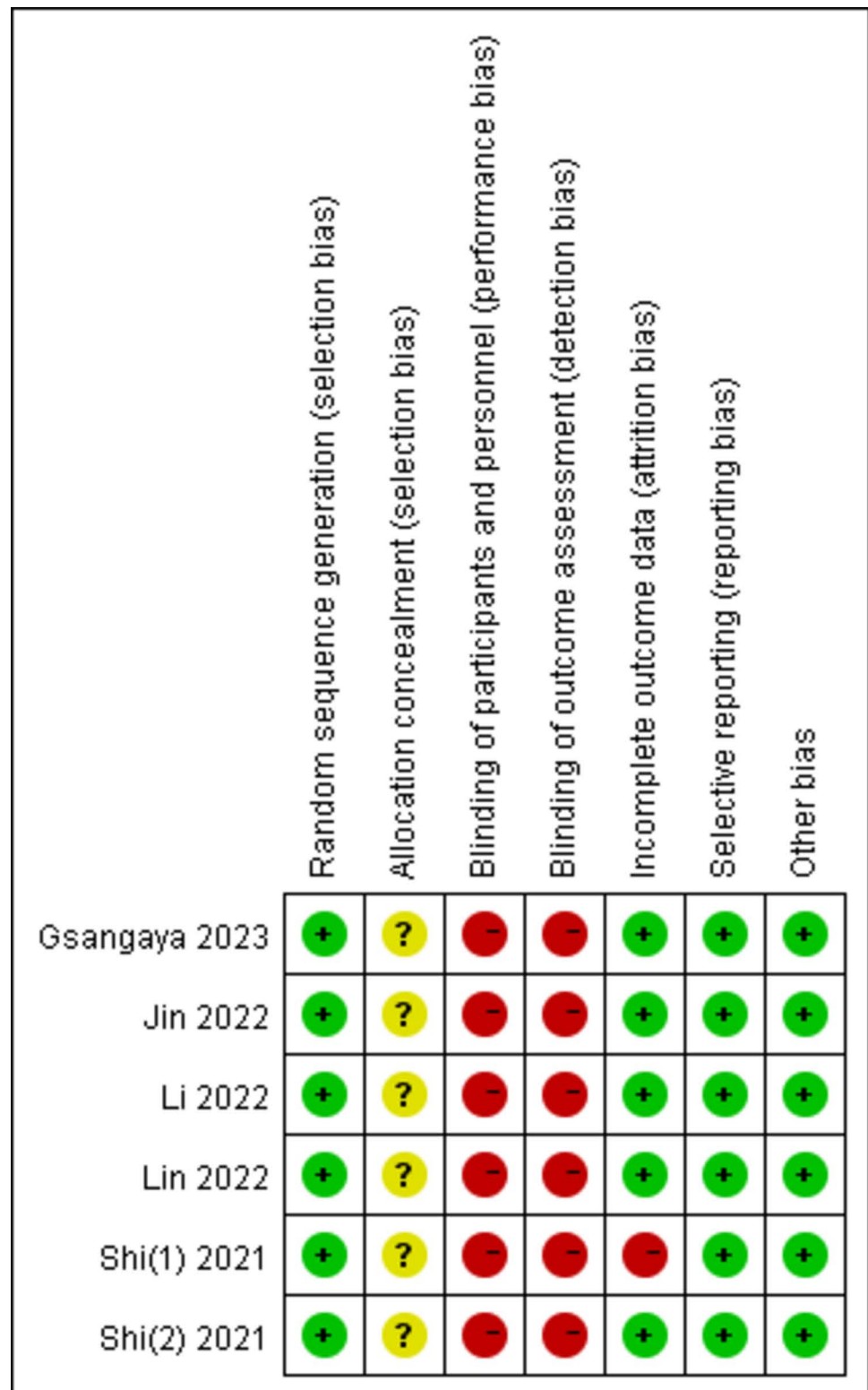

**Fig 3. Risk of bias summary.**

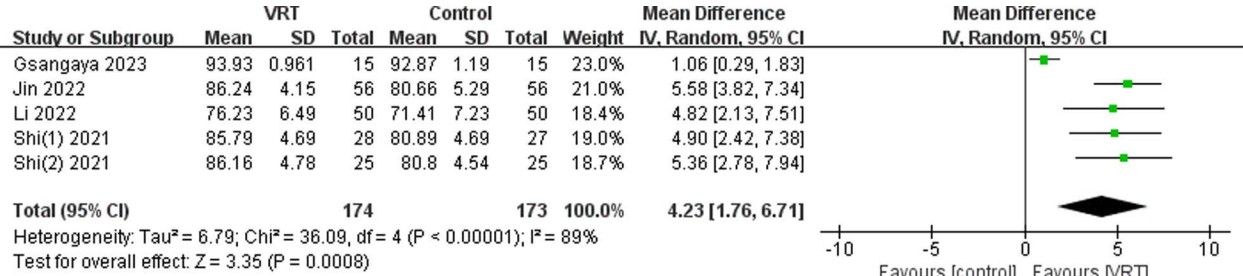

**Fig 4. Forest plot displaying the effects of VRT on IKDC scores.**

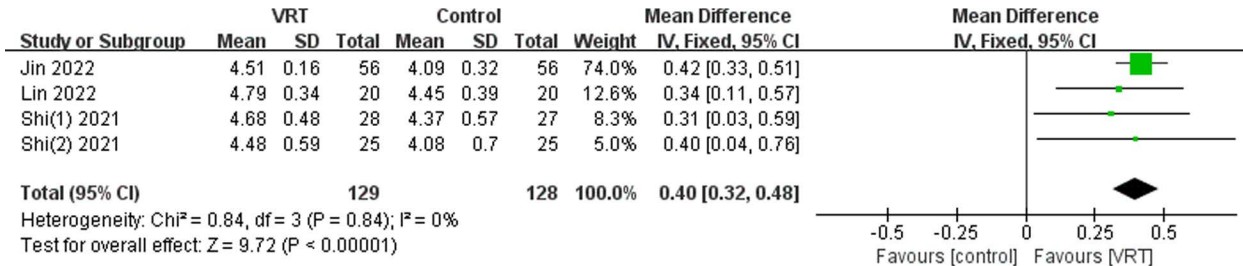

**Fig 5. Forest plot displaying the effects of VRT on FAC scores.**

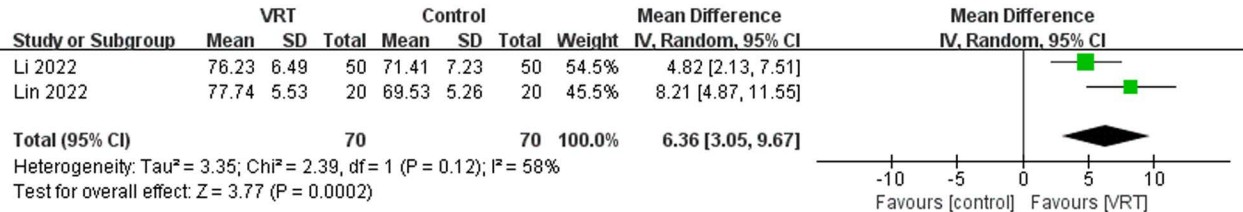

**Fig 6. Forest plot displaying the effects of VRT on Lysholm scores.**

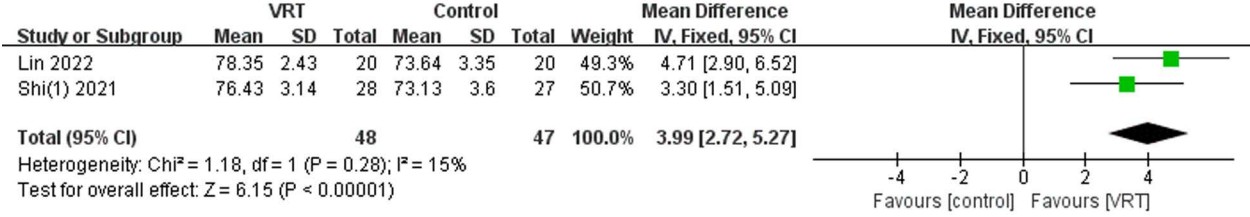

**Fig 7. Forest plot displaying the effects of VRT on step length.**

**Step frequency.** Two studies [27,28] examined the impact of VRT on step frequency in 95 individuals who were undergoing ACLR. Due to significant heterogeneity among studies ($\chi^2$ = 7.53, $P$ = 0.006, $I^2$ = 87%), a random effects model was used for the meta-analysis. The study results indicated a statistically significant variance in step frequency between the two groups (MD: 4.85; 95% CI 0.22-9.47, $P$ = 0.04). Refer to Fig 9 for details.

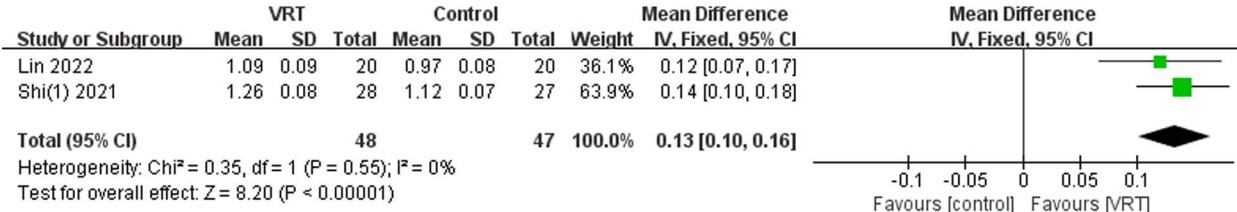

**Fig 8. Forest plot displaying the effects of VRT on step speed.**

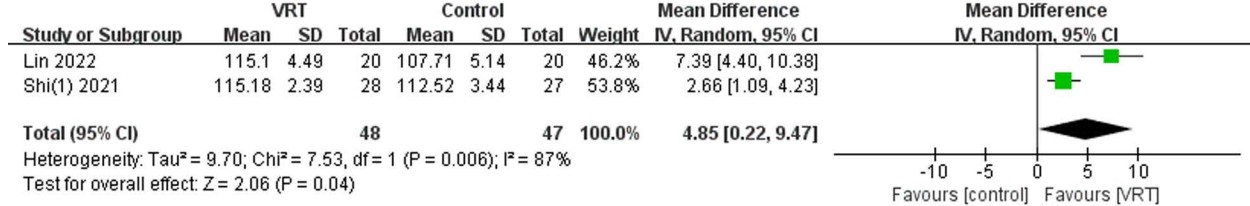

**Fig 9. Forest plot displaying the effects of VRT on step frequency.**

**EPT.** Two studies [25,29] investigated the influence of VRT on EPT in 162 individuals undergoing ACLR. The meta-analysis used a random effects model due to significant heterogeneity among studies ($\chi^2 = 10.86$, $P = 0.001$, $I^2 = 91\%$). The results demonstrated a statistically significant difference in EPT between the two groups (MD: 12.03; 95% CI 3.28-20.78, $P = 0.007$), as shown in Fig 10.

**FPT.** Two studies [25,29] examined the impact of VRT on FPT in 162 patients who were undergoing ACLR. Due to significant heterogeneity among the studies ($\chi^2 = 3.95$, $P = 0.05$, $I^2 = 75\%$), a random effects model was used for the meta-analysis. The study found a statistically significant disparity in FPT between the two groups (MD: 14.57; 95% CI 9.52-19.63, $P < 0.01$), as demonstrated in Fig 11.

## Subgroup analysis

**Immersion level.** Subgroup analyses revealed that knee function was significantly improved with the use of non-immersive VR compared to fully immersive VR (MD: 5.23; 95% CI: 1.76-6.71, $P < 0.00001$). The use of fully immersive VR did not result in any significant improvement in knee function for patients (MD: 3.03; 95% CI -1.17-7.22, $P = 0.16$), as shown in Fig 12.

**Intervention start time.** Subgroup analyses showed that although intervention with VRT started later after surgery (later than 7 days after surgery), it improved knee function (MD: 3.59; 95% CI 0.38-6.80, $P = 0.03$). In addition, starting VRT intervention early after surgery (within 7 days after surgery) resulted in a more significant improvement in knee function (MD: 5.35; 95% CI 3.88-6.83, $P < 0.00001$). Refer to Fig 13.

**Sensitivity analysis.** An analysis was conducted on the six papers included by systematically removing individual studies to assess their impact. The study conducted by Gsangaya et al. [24] revealed a significant contribution to the variability in overall IKDC scores. The removal of this study had a significant impact on heterogeneity ($I^2 = 0\%$), which may be due to the weaker intervention intensity in this study compared with other studies, resulting in a smaller effect size. After removing this study, the fixed-effect model showed a

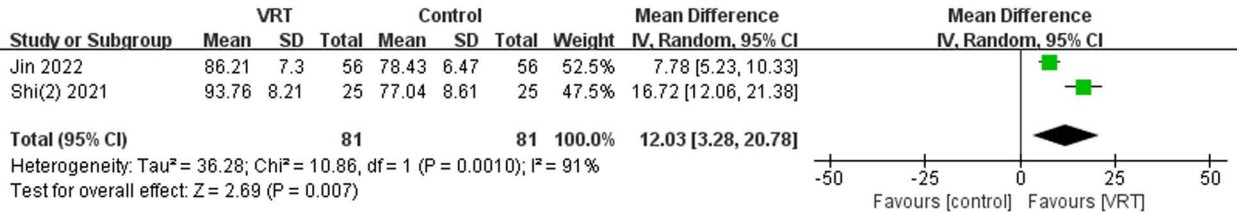

**Fig 10. Forest plot displaying the effects of VRT on EPT.**

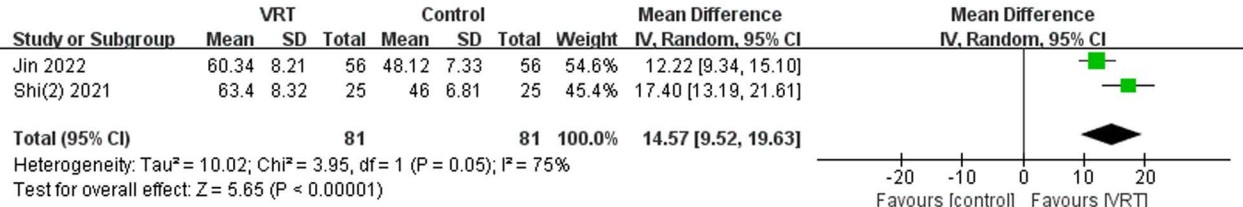

**Fig 11. Forest plot displaying the effects of VRT on FPT.**

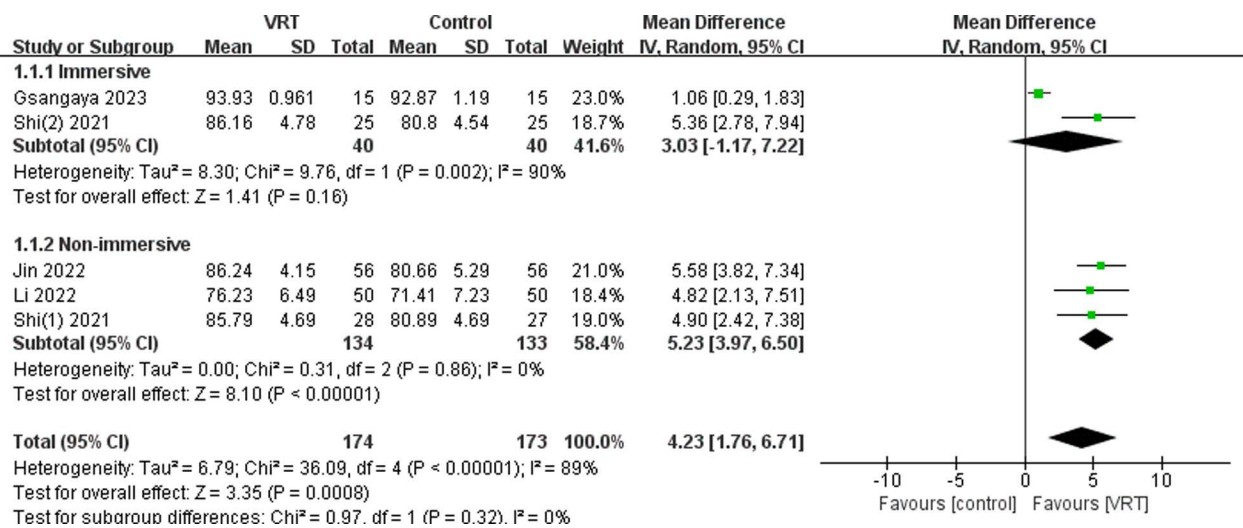

**Fig 12. Subgroup analysis of immersion level by IKDC score.**

statistically significant difference between the two groups (MD: 5.26; 95% CI 4.12-6.40, $P < 0.00001$). The before and after comparisons show little variation, suggesting that the results are consistent and reliable. Please refer to Fig 14.

**Publication bias.** Egger's test was performed for the outcome indicators of 4 or more included studies. The IKDC score was mentioned in five studies [24–26,28,29], and the Egger's test ($P = 0.059$) showed that there was no significant publication bias. The FAC score was mentioned in four studies [25,27–29], and the Egger's test ($P = 0.340$) also showed that there was no significant publication bias.

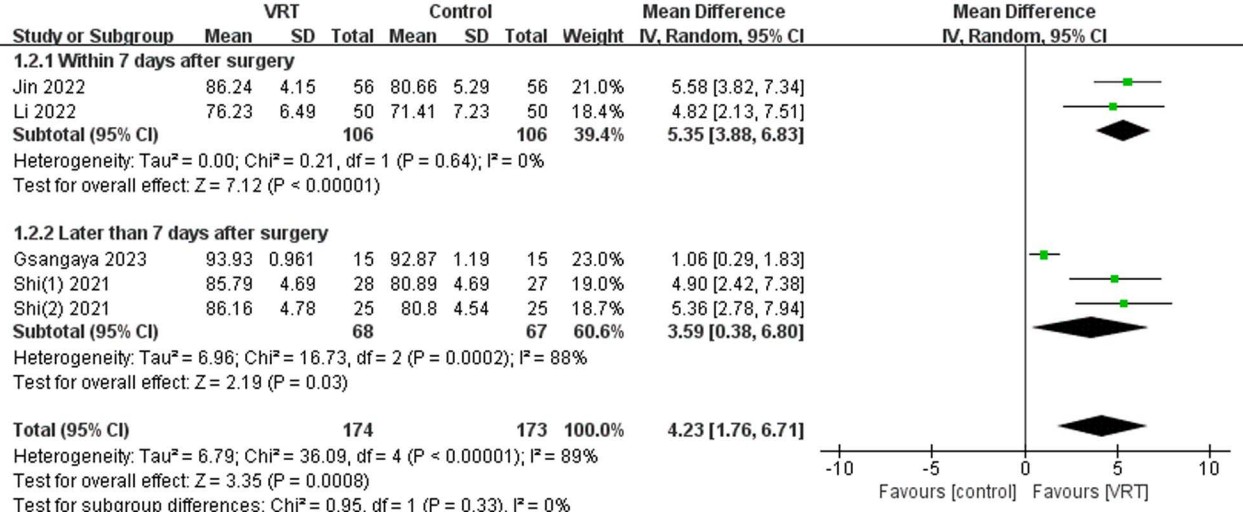

**Fig 13. Subgroup analyses of time to intervention start based on IKDC scores.**

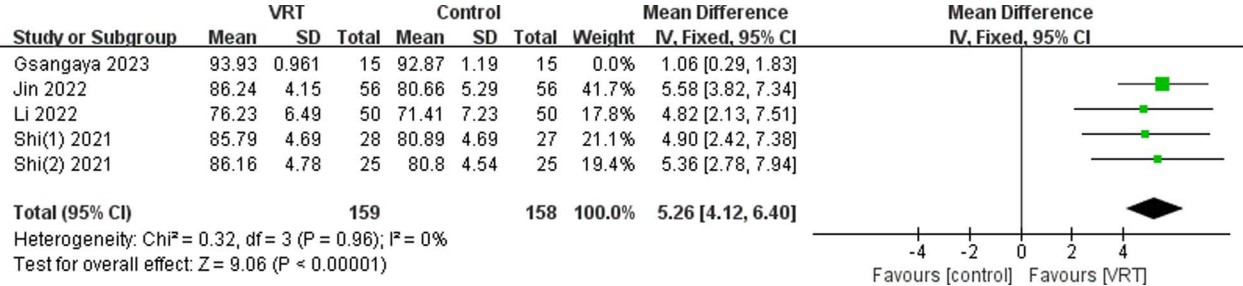

**Fig 14. Sensitivity analysis of the impact of VRT on IKDC scores.**

## Discussion

This systematic review and meta-analysis found for the first time that VRT significantly improved knee function, walking function, gait function, and knee muscle strength in patients after ACLR compared with traditional therapy. Subgroup analysis showed that initiating VRT intervention early postoperatively (within 7 days after surgery) resulted in more significant improvements in knee function compared to late postoperatively (later than 7 days after surgery). For the level of immersion, subgroup analysis showed that non-immersive VR resulted in more significant improvements in knee function than immersive VR. However, the reports by D'Ambrosi et al. [30,31] suggested a poor effect, which may be mainly due to the lack of immersion and interactivity of educational videos on TikTok and YouTube. VR immerses patients by simulating multi-sensory integrated scenarios in which they can interact. This reduces the patient's fear of movement and increases their motor output. The absence of mechanoreceptors after ACLR reduces proprioceptive signals to the brain [32], which can lead to reduced proprioception. Improved proprioception contributes to the knee's functional and dynamic stability [33]. This is one of the potential explanations for the systematic review's finding that VRT outperformed traditional therapy in enhancing knee function, specifically by enhancing the patient's proprioceptive function through multisensory stimulation [34,35].

Another reason may be that VRT improves knee stability and thus knee function through improvements in periprosthetic strength. The review by Song et al. [36] is consistent with our findings. They reported that VR represents a promising approach to effectively improve joint function. However, in a RCTs by Karakoc et al. [20], no difference was found between VRT and conventional therapy. This may be due to the fact that one of the inclusion criteria for that study was patients who had undergone arthroscopic ACLR within the last six months, and most patients have generally begun to gradually return to normal life and sports at six months postoperatively, by which time knee function has almost always returned to normal.

For walking ability, VRT is superior to traditional therapy probably because it improves walking ability by enhancing knee stability and joint mobility in ACLR patients. In VR, interactive training experiences and challenging training modes allow patients to better respond to different virtual environments in real time. Moreover, patients immersed in safe and intriguing virtual environments can greatly reduce their internal focus on themselves, thus reducing exercise fear [19]. These virtual environments not only boost patients' motivation and collaboration during walking exercises, but also enhance their trunk stability and precise control of knee movements, leading to a significant improvement in their walking function. A previous meta-analysis by Hao et al. [37] yielded similar results, and their study concluded that VRT was more effective in training walking ability compared to conventional therapy.

Muscle strength, proprioception, and balance are important factors affecting gait [38,39]. Therefore, the reasons why VRT can effectively improve gait function may be as follows. VRT can simulate various life imbalances, and patients can control their bodies to make appropriate postures according to specific tasks. This measure results in more precise, rapid, and effective activation of the patient's knee muscle groups, and the stimulation of different sensory information therein also helps to improve sensory processing and sensorimotor coordination [40]. Results consistent with ours were also obtained in a systematic review that included 87 RCTs by Zhang et al. [41], who found that VRT had a greater effect on improving gait function compared to traditional therapy. However, the systematic review by Moreno-Verdu et al. [42] found that VRT was not superior to conventional therapy. The reason for this may be that the training programs in some of the included studies did not focus on gait training, while some studies were also limited by the VR equipment and did not make good use of VR.

Insufficient muscle strength around the knee joint in patients after ACLR can lead to decreased knee function, altered knee biomechanics, and even secondary injury to the ACL [43]. The reasons why VRT is superior to traditional therapy in improving muscle strength may be twofold. On the one hand, the selected virtual games contain more lower limb hip and knee muscle strength training game modules, which require gross movements of the limbs and continuous postural changes in order to complete the tasks and can effectively enhance the patients' muscle strength [19]. On the other hand, VR scenarios can help patients improve their motor imaginative skills and help the central nervous system integrate proprioception through visual, auditory, and haptic feedback to create joint positional and kinaesthetic senses. This changes neuromuscular excitation and improves motor output. A previous systematic review by Wei et al. [44] also yielded consistent results, as they found that VRT improved knee muscle strength more than traditional therapy and was more favourable to improvements in flexor strength. However, a study by Baltaci et al. [45] found no significant difference in knee muscle strength improvement between the VR group and the control group after a 12-week intervention in patients undergoing ACLR. This may be due to the fact that boxing and bowling in VR games do not work well with lower limb function.

Subgroup analysis found that non-immersive VR improved knee function better than immersive, which is consistent with the results obtained by Wang et al. [46]. Possible explanations for this include the fact that non-immersive VR maintains patients' connection to

reality, and when compared to immersive VR, it causes fewer side effects like dizziness, nausea, and vomiting, making it more suitable for patients with weaker psychological constructs [47]. We found that starting the intervention with VRT early in the post-surgical period (within 7 days after surgery) resulted in a more significant improvement in knee function. This suggests that future studies could develop a reasonable rehabilitation program to start the intervention as early as possible in order to achieve better rehabilitation results. The sensitivity analysis found significantly less heterogeneity, excluding Gsangaya et al. [24]. This may be due to the fact that the study was a fortnightly intervention, whereas the other studies had a minimum of five interventions a week, and the lack of intensity of the intervention resulted in a lower effect size.

After ACL rupture, the brain will rely more on visual cognition to help the body maintain stability [48]. This provides excellent conditions for the use of VRT in post-ACLR rehabilitation, and as the cost of VRT decreases, VRT will become a favourable tool in post-ACLR rehabilitation. VRT can provide a safe environment that allows patients to reduce the risk of injury. It also reduces the patient's intrinsic preoccupation, allowing them to perform bedside as well as out-of-room activities. Furthermore, it improves patients' recovery speed, reducing hospitalization costs.

## Limitations

To the best of our knowledge, this is the first systematic review and meta-analysis of the effectiveness of VRT in rehabilitation after ACLR. However, some limitations affecting the results should be noted. (1) We included a limited number of trials, which would have limited the ability to detect small effect sizes. (2) The quality, number, and sample size of RCTs included were far from ideal. Despite our efforts to contact authors for data, some studies were excluded due to a lack of analysis data. Furthermore, blindness was not mentioned in any of the studies. Thus, the conclusions of this systematic evaluation should be treated with caution. (3) The results of this review may have been influenced by variations in the duration and type of VRT used to rehabilitate patients, as well as the use of VR equipment and models. So, we encourage researchers to do higher-quality RCTs with larger sample sizes and to standardise VRT intervention protocols so that they can find the most important parts of intervention that help people recover the most after ACLR. (4) Surgical approaches and types of grafts were not standardised in the included literature, although current evidence does not show differences in biomechanical outcomes between graft types [49,50]. We look forward to more future studies that support the impact of surgical approaches and grafts on postoperative rehabilitation in patients undergoing ACLR.

## Conclusion

In summary, VRT has been shown to be more effective than traditional rehabilitation in improving knee function, walking function, gait function, and knee muscle strength in patients after ACLR. Non-immersive VR was found to be more efficacious in enhancing knee function compared to completely immersive VR. Improvement of knee function was more significant when the intervention was started early in the postoperative period (within 7 days after surgery) than later in the period (after 7 days after surgery). This innovative and interesting approach contributes to the rehabilitation of patients undergoing ACLR. However, most of the existing research on VR for ACLR relies on visual, auditory, and tactile inputs to help patients train, whereas proprioception includes a variety of senses such as visual, tactile, positional, and kinesthetic senses. Therefore, future studies should combine other methods or develop new VRT solutions to increase the input of other senses. However, due to the limited

number and quality of the studies included, it is necessary to conduct high-quality multi-centre, large-sample, randomized, double-blind, controlled clinical trials in the future. These trials will help further validate the effectiveness of VRT in rehabilitating patients after ACLR. Moreover, the optimal duration range, frequency, and intensity of VRT rehabilitation training should be continuously explored to standardise its standardised rehabilitation process. Thereby, a more effective and affordable rehabilitation intervention programme should be provided for the rehabilitation of patients after ACLR.

## Supporting Information

**S1 Checklist. PRISMA 2020 checklist.**
(DOCX)

**S1 File. Search strategies.**
(PDF)

**S1 Table. All studies identified in the literature search.**
(XLSX)

**S2 Table. Data.**
(XLSX)

**S3 Table. Cochrane risk assessment.**
(XLSX)

## Author contributions

**Data curation:** Yunchuan Li, Junjie Peng.

**Formal analysis:** Yunchuan Li, Jintao Cao.

**Funding acquisition:** Jintao Cao, Feng'e Qian.

**Investigation:** Yunchuan Li, Yang Ou, Weisha Ma.

**Methodology:** Yunchuan Li, Jiaming Wu, Weisha Ma.

**Project administration:** Weisha Ma, Xiaoqian Li.

**Supervision:** Feng'e Qian, Xiaoqian Li.

**Validation:** Yunchuan Li, Jiaming Wu.

**Visualization:** Yunchuan Li, Yang Ou.

**Writing – original draft:** Yunchuan Li.

**Writing – review & editing:** Yunchuan Li, Junjie Peng, Feng'e Qian.

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
