## [Decision Letter · Decision Letter 0]

20 Aug 2024

PONE-D-24-25139Effects of virtual reality technology on anterior cruciate ligament reconstruction: A systematic review and meta-analysisPLOS ONE

Dear Dr. Qian,

Thank you for submitting your manuscript to PLOS ONE. After careful consideration, we feel that it has merit but does not fully meet PLOS ONE’s publication criteria as it currently stands. Therefore, we invite you to submit a revised version of the manuscript that addresses the points raised during the review process.

=Please submit your revised manuscript by Oct 04 2024 11:59PM. If you will need more time than this to complete your revisions, please reply to this message or contact the journal office at plosone@plos.org . Please include the following items when submitting your revised manuscript:

We look forward to receiving your revised manuscript.

Kind regards,

Luciana Labanca

Academic Editor

PLOS ONE

Journal Requirements:

2. Thank you for stating the following financial disclosure: Science and Technology Programme of Yunnan Provincial Science and Technology Department (Grant NO. 202301AZ070001-052), Scientific Research Fund Project of Yunnan Provincial Department of Education (Grant NO. 2023Y0493).  

3. Please include your tables as part of your main manuscript and remove the individual files. Please note that supplementary tables (should remain/ be uploaded) as separate ""supporting information"" files

Reviewers' comments:

Reviewer's Responses to Questions

**Comments to the Author**

1. Is the manuscript technically sound, and do the data support the conclusions?

Reviewer #1: Yes

Reviewer #2: Partly

2. Has the statistical analysis been performed appropriately and rigorously? 

Reviewer #1: Yes

Reviewer #2: Yes

3. Have the authors made all data underlying the findings in their manuscript fully available?

Reviewer #1: Yes

Reviewer #2: No

4. Is the manuscript presented in an intelligible fashion and written in standard English?

Reviewer #1: No

Reviewer #2: Yes

5. Review Comments to the Author

Reviewer #1: The manuscript presents a systematic review and meta-analysis of the effects of virtual reality technology (VRT) on anterior cruciate ligament (ACL) reconstruction rehabilitation. The study design, objectives, and methodology are clearly stated, and the results section provides a summary of the findings. However, there are several areas where the manuscript could be improved to enhance clarity, precision, and adherence to best practices in scientific reporting.

1. The abstract should succinctly summarize the novel findings and their implications, such as “This study is the first to systematically compare VRT with traditional therapy, providing evidence for the integration of VRT in post-ACLR rehabilitation protocols.”

2. The introduction should clearly state the gap in knowledge that the study addresses. Add a sentence that highlights the lack of consensus on the use of VRT in ACLR rehabilitation, emphasizing the study's contribution to filling this gap.

3. Clarify the time frames mentioned in this section. It is unclear what is meant by “more than one week” and “one week after surgery.” Specify the exact time points to avoid ambiguity.

4. The sensitivity analysis is well-conducted. However, it would be helpful to explain why the study by Gsangaya et al. had such a significant impact on the overall results. Consider adding a sentence or two discussing possible reasons.

5. The publication bias section is concise. However, it would be useful to mention the number of studies included in the Egger's test and the effect size.

6. The discussion should provide a more comprehensive synthesis of the findings in relation to existing literature. Discuss the mechanisms behind the observed effects, the clinical implications, and potential future research directions. Also, consider mentioning the limitations of the study and how they might affect the interpretation of the results.

7. Some of the statements in the manuscript are not clearly expressed and the quality of the language needs to be improved. Such as:

Current: This study aimed to examine the effects of VRT on rehabilitation following ACLR. The findings will help inform decisions regarding the use of VRT in clinical settings.

Revision Suggestion: This study aimed to evaluate the effects of virtual reality technology on rehabilitation following ACLR, providing insights for its application in clinical settings.

Current: After a selection process, six randomised controlled studies including a total of 387 patients who underwent ACL repair were considered.

Revision Suggestion: Following a rigorous selection process, six randomized controlled trials involving a total of 387 patients who underwent ACL reconstruction were included.

Current: The results indicate that VRT had a significant positive effect on various measures.

Revision Suggestion: The findings demonstrate that VRT significantly enhances knee function, walking ability, gait function, and knee muscular strength post-ACLR.

8. Ensure consistent use of terms (e.g., always use “virtual reality technology (VRT)” after its first mention).

9. Ensure that all references cited in the text are included in the reference list and vice versa. Double-check the formatting of the references to ensure consistency and adherence to the journal's style guide.

Reviewer #2: Title: really poor and scarce

Abstract

Report clear purpose and hypothesis if any

Design: ok

Objectives:any hypothesis

Methods: report keywords added

Results: too many numbers. hard to read

Conclusions: coherent

Introduction

Little bit too long

Please focus on the topic of your paper

Report current evidences and what are the controversies in literature

Explain the rationale for your study

finish with aim and hypothesis

methods

improve inclusion and exclusion criteria

results

little bit too long

discussion

start with main findings of your paper

report what is new and what this paper add to Current literature

explain how this study can help in clinical daily practice

improve limitations

conclusions

coherent

references

add following

D'Ambrosi R, Hewett TE. Validity of Material Related to the Anterior Cruciate Ligament on TikTok. Orthop J Sports Med. 2024 Feb 22;12(2):23259671241228543. doi: 10.1177/23259671241228543. PMID: 38405012; PMCID: PMC10893838.

D'Ambrosi R, Milinkovic DD, Abermann E, Herbort M, Fink C. Quality of YouTube Videos Regarding Anterior Cruciate Ligament Reconstruction Using Quadriceps Tendon Autograft Is Unsatisfactory. Arthroscopy. 2024 Aug;40(8):2236-2243. doi: 10.1016/j.arthro.2024.01.002. Epub 2024 Jan 6. PMID: 38185185.

6. PLOS authors have the option to publish the peer review history of their article (what does this mean? ). If published, this will include your full peer review and any attached files.

**Do you want your identity to be public for this peer review?** For information about this choice, including consent withdrawal, please see our Privacy Policy .

Reviewer #1: No

Reviewer #2: No

---

## [Author Response · Author response to Decision Letter 1]

4 Oct 2024

For Reviewer #1

Comment 1: The abstract should succinctly summarize the novel findings and their implications, such as “This study is the first to systematically compare VRT with traditional therapy, providing evidence for the integration of VRT in post-ACLR rehabilitation protocols.”

Response: Thank you very much for your comment. We have changed the conclusion section of the abstract to the following. “This study is the first to systematically compare VRT with traditional therapy, providing evidence for the integration of VRT in post-ACLR rehabilitation protocols. VRT might be an effective and promising therapy in rehabilitation after ACLR. However, more high-quality studies with large samples are needed to verify the findings”.

Comment 2: The introduction should clearly state the gap in knowledge that the study addresses. Add a sentence that highlights the lack of consensus on the use of VRT in ACLR rehabilitation, emphasizing the study's contribution to filling this gap.

Response: Thank you very much for your comment. We have made the following changes to the introduction. “In addition, there is a lack of consensus regarding the use of VRT in ACLR rehabilitation. Therefore, clarifying whether and to what extent VRT is effective for rehabilitation after ACLR is important for future research and patient treatment choice.

Based on this background, we hypothesised that a systematic review and meta-analysis of randomised controlled trials (RCTs) would provide sufficient scientific evidence that VRT is an effective therapy for rehabilitation after ACLR. Therefore, the aim of this study was to comprehensively evaluate the efficacy of VRT compared with conventional therapies in rehabilitation after ACLR, which will provide a basis for decision-making on its clinical application and evidence for consensus building.”

Comment 3: Clarify the time frames mentioned in this section. It is unclear what is meant by “more than one week” and “one week after surgery.” Specify the exact time points to avoid ambiguity.

Response: We really appreciate your efforts and comments on our manuscript. Time frames have been specified in the subgroup analysis section. Specifically, “later than 7 days after surgery” and “within 7 days after surgery”.

Comment 4: The sensitivity analysis is well-conducted. However, it would be helpful to explain why the study by Gsangaya et al. had such a significant impact on the overall results. Consider adding a sentence or two discussing possible reasons.

Response: Thank you very much for your comment. Possible causes have been added, as shown below. “The removal of this study had a significant impact on heterogeneity (I2 = 0%), which may be due to the weaker intervention intensity in this study compared with other studies, resulting in a smaller effect size.”

Comment 5: The publication bias section is concise. However, it would be useful to mention the number of studies included in the Egger's test and the effect size.

Response: Thank you very much for your comment. The number of studies included in the Egger's test and the effect size have been mentioned. As shown in the following. “The IKDC score was mentioned in five studies [24-26,28,29], and the Egger's test (P = 0.059) showed that there was no significant publication bias. The FAC score was mentioned in four studies [25,27-29], and the Egger's test (P = 0.340) also showed that there was no significant publication bias.”

Comment 6: The discussion should provide a more comprehensive synthesis of the findings in relation to existing literature. Discuss the mechanisms behind the observed effects, the clinical implications, and potential future research directions. Also, consider mentioning the limitations of the study and how they might affect the interpretation of the results.

Response: We really appreciate your efforts and comments on our manuscript. A more comprehensive review of the findings has been provided in the context of the existing literature. Mechanisms behind the observed effects, clinical significance and potential future research directions have been discussed. Limitations of the study and how these limitations affect the interpretation of the results have been mentioned. See both the Discussion and Limitations sections of the manuscript for more details.

Comment 7: Some of the statements in the manuscript are not clearly expressed and the quality of the language needs to be improved. Such as: Current: This study aimed to examine the effects of VRT on rehabilitation following ACLR. The findings will help inform decisions regarding the use of VRT in clinical settings.

Current: This study aimed to examine the effects of VRT on rehabilitation following ACLR. The findings will help inform decisions regarding the use of VRT in clinical settings.

Revision Suggestion: This study aimed to evaluate the effects of virtual reality technology on rehabilitation following ACLR, providing insights for its application in clinical settings.

Response: Thank you very much for your comment. This section has been improved. As shown in the following. “We hypothesised that VRT would be a more effective treatment than traditional therapy in post-ACLR rehabilitation. This study aimed to evaluate the effects of VRT on rehabilitation following ACLR, providing insights for its application in clinical settings.”

Current: After a selection process, six randomised controlled studies including a total of 387 patients who underwent ACL repair were considered.

Revision Suggestion: Following a rigorous selection process, six randomized controlled trials involving a total of 387 patients who underwent ACL reconstruction were included.

Response: Thank you very much for your comment. This section has been improved. As shown in the following. “Following a rigorous selection process, six RCTs involving a total of 387 patients who underwent ACL reconstruction were included.”

Current: The results indicate that VRT had a significant positive effect on various measures.

Revision Suggestion: The findings demonstrate that VRT significantly enhances knee function, walking ability, gait function, and knee muscular strength post-ACLR.

Response: Thank you very much for your comment. This section has been improved. As shown in the following. “The findings demonstrated that VRT significantly enhanced knee function, walking ability, gait function, and knee muscle strength post-ACLR.”

Comment 8: Ensure consistent use of terms (e.g., always use “virtual reality technology (VRT)” after its first mention).

Response: Thank you very much for your comment. Consistency in the use of terminology has been ensured. Checks have been made to ensure that“acronyms” are always used after the first reference.

Comment 9: Ensure that all references cited in the text are included in the reference list and vice versa. Double-check the formatting of the references to ensure consistency and adherence to the journal's style guide.

Response: Thank you very much for your comment. All cited references have been carefully checked for number, format, and style.

For Reviewer #2

Comment 1: Title: really poor and scarce

Response: Thank you very much for your comment. The title has been changed from "Effects of virtual reality technology on anterior cruciate ligament reconstruction: A systematic review and meta-analysis" to "Effectiveness of virtual reality technology in rehabilitation after anterior cruciate ligament reconstruction: A systematic review and meta-analysis".

Comment 2: Methods: report keywords added.

Response: Thank you very much for your comment. We have added the keyword in the Methods. As shown in the following. “The literature search was conducted from the inception of the database to March 2024, utilizing keywords such as "anterior cruciate ligament", "anterior cruciate ligament reconstruction", "anterior cruciate ligament injury", and "virtual reality".”

Comment 3: Objectives: any hypothesis

Response: Thank you very much for your comment. We have added the hypothesis in the Objectives section. As shown in the following. “We hypothesised that VRT would be a more effective treatment than traditional therapy in post-ACLR rehabilitation.”

Comment 4: Results: too many numbers. hard to read.

Response: Thank you very much for your comment. Appropriate deletions have been made to the figures in the Results section. Please see manuscript for details.

Comment 5: Introduction: little bit too long. Please focus on the topic of your paper.

Response: We really appreciate your efforts and comments on our manuscript. The Introduction has been appropriately restructured by focusing on the theme. For details, see the foreword in the manuscript.

Comment 6: Report current evidences and what are the controversies in literature.

Response: We really appreciate your efforts and comments on our manuscript. Current evidence has been reported as well as controversies in the literature. Please see manuscript for details.

Comment 7: Explain the rationale for your study.

Response: Thank you very much for your comment. The fundamentals of virtual reality technology have been mentioned. As shown in the following. “VR is a digital simulation of a computer-generated situation or environment that generates a realistic environment for task-specific training in which the user can orientate and interact in 3D through multiple sensory modalities].”

Comment 8: Finish with aim and hypothesis.

Response: Thank you very much for your comment. We have modified the end of the introduction to include the aims and hypothesis in the content. As shown in the following. “Based on this background, we hypothesised that a systematic review and meta-analysis of randomised controlled trials (RCTs) would provide sufficient scientific evidence that VRT is an effective therapy for rehabilitation after ACLR. Therefore, the aim of this study was to comprehensively evaluate the efficacy of VRT compared with conventional therapies in rehabilitation after ACLR, which will provide a basis for decision-making on its clinical application and evidence for consensus building.”

Comment 9: Improve inclusion and exclusion criteria.

Response: We really appreciate your efforts and comments on our manuscript. We have improved the inclusion and exclusion criteria. Please see manuscript for details.

Comment 11: Results: little bit too long.

Response: Thank you very much for your comment. Appropriate adjustments have been made to the Results section to ensure what must be in this study. See the Results section of the manuscript for details.

Comment 10: Discussion: start with main findings of your paper.

Response: Thank you very much for your comment. The Discussion section has been modified to report the main findings of this study from the beginning. Please see manuscript for details.

Comment 11: Report what is new and what this paper add to Current literature.

Response: We really appreciate your efforts and comments on our manuscript. Because this is the first systematic review and meta-analysis of the effectiveness of VRT in post-ACLR rehabilitation. Therefore, this article adds some directions for future research that could be conducted with this patient using VRT. As shown in the following. “However, most of the existing research on VR for ACLR relies on visual, auditory, and tactile inputs to help patients train, whereas proprioception includes a variety of senses such as visual, tactile, positional, and kinesthetic senses. Therefore, future studies should combine other methods or develop new VRT solutions to increase the input of other senses.”

Comment 12: Explain how this study can help in clinical daily practice.

Response: Thank you very much for your comment. An explanation of how this research can help clinical daily practice has been included in the article. As shown in the following. “VRT can provide a safe environment that allows patients to reduce the risk of injury. It also reduces the patient's intrinsic preoccupation, allowing them to perform bedside as well as out-of-room activities. Furthermore, it improves patients' recovery speed, reducing hospitalization costs.”

Comment 13: Improve limitations.

Response: Thank you very much for your comment. Improvements have been made to the Limitations section. Please see manuscript for details.

Comment 13: References: add following.

Response: Thank you very much for your comment. We have added these two literatures in the Discussion section, in the text as [30,31]. Please see manuscript for details.

---

## [Decision Letter · Decision Letter 1]

18 Nov 2024

Effectiveness of virtual reality technology in rehabilitation after anterior cruciate ligament reconstruction: A systematic review and meta-analysis

PONE-D-24-25139R1

Dear Dr. Qian,

We’re pleased to inform you that your manuscript has been judged scientifically suitable for publication and will be formally accepted for publication once it meets all outstanding technical requirements.

Kind regards,

Luciana Labanca

Academic Editor

PLOS ONE

Additional Editor Comments (optional):

Reviewers' comments:

Reviewer's Responses to Questions

**Comments to the Author**

1. If the authors have adequately addressed your comments raised in a previous round of review and you feel that this manuscript is now acceptable for publication, you may indicate that here to bypass the “Comments to the Author” section, enter your conflict of interest statement in the “Confidential to Editor” section, and submit your "Accept" recommendation.

Reviewer #1: All comments have been addressed

Reviewer #2: All comments have been addressed

2. Is the manuscript technically sound, and do the data support the conclusions?

Reviewer #1: Yes

Reviewer #2: Yes

3. Has the statistical analysis been performed appropriately and rigorously? 

Reviewer #1: Yes

Reviewer #2: Yes

4. Have the authors made all data underlying the findings in their manuscript fully available?

Reviewer #1: Yes

Reviewer #2: Yes

5. Is the manuscript presented in an intelligible fashion and written in standard English?

Reviewer #1: Yes

Reviewer #2: Yes

6. Review Comments to the Author

Reviewer #1: (No Response)

Reviewer #2: all queries have been answered in full so the article deserve to be considered for publication in the journal

7. PLOS authors have the option to publish the peer review history of their article (what does this mean? ). If published, this will include your full peer review and any attached files.

**Do you want your identity to be public for this peer review?** For information about this choice, including consent withdrawal, please see our Privacy Policy .

Reviewer #1: No

Reviewer #2: No

---

## [Editor Report · Acceptance letter]

PONE-D-24-25139R1

PLOS ONE

Dear Dr. Qian,

I'm pleased to inform you that your manuscript has been deemed suitable for publication in PLOS ONE. Congratulations! Your manuscript is now being handed over to our production team.

Kind regards,

on behalf of

Dr. Luciana Labanca

Academic Editor

PLOS ONE